# Traffic Safety Improvement via Optimizing Light Environment in Highway Tunnels

**DOI:** 10.3390/ijerph19148517

**Published:** 2022-07-12

**Authors:** Baofeng Su, Jiangbi Hu, Juncheng Zeng, Ronghua Wang

**Affiliations:** 1Faculty of Architecture, Civil and Transportation Engineering, Beijing University of Technology, Beijing 100124, China; bf12087121@hotmail.com (B.S.); wangrh@bjut.edu.cn (R.W.); 2Fujian Expressway Science & Technology Innovation Research Institute Co., Ltd., Fuzhou 350001, China; ulzeng@126.com

**Keywords:** light environment, tunnel lighting, visual health, traffic safety, intelligent control

## Abstract

Driving in tunnel areas depends more heavily on light conditions than that on open roadways. Traditional lighting systems in highway tunnels adjust lighting parameters only caring about outside light luminance, and focus is usually on energy conservation; however, little concern is about drivers’ actual physical and psychological needs. How to leverage the enormous research progress of traffic safety, light environment, human factors engineering, and modern lighting sources to create an ideal tunnel light environment that aids with ensuring driving safety and lower interference effects caused by the change of light environment will greatly improve safety level and reduce adverse influence on drivers’ visual health in a tunnel area. An intelligent lighting control system designed with multiple influence factors are systematically considered. Based on sensor data from outside natural light conditions, target lighting parameters are determined per each lighting zone requires; then, lighting commands will be transferred and parsed by adaptive lighting controllers and modules, eventually LED lighting properties are altered step by step. This system helps a lot with optimizing tunnel lighting quality and improving drivers’ visual performance; as a result, it contributes to lower the fluctuation of drivers’ workload and get a smooth traffic flow, and ultimately this technically ensures physical and mental health of drivers in a tunnel area.

## 1. Background

In recent years, highway tunnel construction is increasing on a global scale. As of 2020, China had 21,316 highway tunnels or 21.9993 million meters, and the scale and number rank first in the world [1]. Table 1 also shows that, from 2011 to 2020, the proportion of long and extra-long tunnels rose year by year from 17.6% and 3.8% to 26% and 6.5%, respectively, and annual growth rates of both types exceeded that of total tunnels.

While improving traffic efficiency, to some extent, tunnels are also possible bottleneck sections of highway network. 2019 annual statistical data of Traffic Management Bureau of Ministry of Public Security of China indicate that the number of casualties caused by tunnel accidents accounts for 0.32% of total casualties caused by road traffic accidents, and direct property loss is about 0.95% of total loss, although tunnel traffic accidents occupy only 0.23% of total count of road traffic accidents [2]. Meanwhile, because tunnel space is relatively closed and there are wide variances between tunnel area and open roadways, this leads to increasing traffic risks in the tunnel area, and tunnel traffic accidents are characterized by complex causes, short disaster time, inconvenient evacuation, and difficult rescue. Moreover, once such a traffic accident occurs, if protection measures are inappropriate, it is easy to cause second accidents resulting in aggravated chain reactions, such as rescue difficulty, economic loss, social impact, etc.

How to reduce frequency and extent of injury of tunnel traffic accidents has motivated a widespread concern because this directly affects whether tunnel construction and operation can really improve traffic efficiency. Road traffic accidents are caused by synergic effects of people, vehicle, road, and environment. As denoted in Figure 1, traffic accident causes are grouped into three major categories in America, driver related factors dominate, and reliability of drivers has a decisive influence on traffic safety [3]. During the driving process, the road environment greatly affects drivers’ cognition, decision-making and behavior, so road environment is a potentially important factor affecting traffic safety.

Studies show that the major source for drivers to sense road conditions is vision under normal driving conditions [4], and drivers rely on vision to cognize more than 80% of road traffic information in the surroundings [5]. Drivers’ visual condition is closely related to traffic safety. The optical environment in highway tunnels directly influences a lot on visual characteristics of drivers. Highway tunnel is a special road structure which makes internal space semi-isolated from the external environment with entrances or exits as the transition zone between different traffic environments. The inherent nature of tunnel structure makes it a very special road traffic environment, which is characterized by relatively narrow and almost closed space, clear distinction of lighting environment between inside and outside space, low luminance inside the tunnel, slow diffusion of smoke and dust, etc. Driving inside tunnels with bad lighting conditions for a long time will inevitably induce fear, irritability, depression, visual fatigue, and other adverse physiological and psychological reactions, which have a negative effect on driving safety and can easily cause traffic accidents. Research data show about 8% of skilled drivers and 14% of drivers first driving through the tunnel area feel depressed while driving inside [6].

In order to investigate the rationality of light environment of the tunnels in service of a certain province in China, the research team designed a questionnaire for drivers and conducted a large number of questionnaire surveys on the overall lighting conditions of more than 10 highway tunnels. Statistical results indicate that, for all drivers that participated in this survey, 41% feel that overall tunnel lighting conditions are rather dark and 14% feel lighting in the interior zone is too dark as Figure 2, 78% think exit and threshold zones are the most dangerous tunnel zones as Figure 3, and 61% consider 11:00 A.M. to 1:00 P.M. of daytime and nighttime are unsafe periods for tunnel driving as Figure 4.

The above study shows that apparent discrepancies of lighting environment existing in entrance or exit and low luminance in interior zone are both major causes resulting in degradation of drivers’ visual ability and cognition, eventually to trigger traffic accidents more frequently. However, it is uneconomical and technically unpractical to maintain exactly the same illumination level for the tunnel light environment as outside natural light.

Based on the above analysis of actual tunnel lighting problems, firstly this study is to sum up the theoretical bases, such as light environment theories of highway tunnel, safety and comfort requirements on driving vision in tunnel environment, characteristics and applications of tunnel light sources, and research achievements of tunnel lighting control systems. Then, this research will explore and design an innovative tunnel lighting control system which is able to meet drivers’ vision and cognition demand, to operate with low energy consumption, to make artificial and natural light sources friendly coupled, and eventually to create comfortable conditions to improve driving safety level in the tunnel area.

## 2. Traffic Safety and Light Environment in Highway Tunnels

### 2.1. Light Environment in Tunnels: Definition and Objective

Tunnel light environment is the dynamically changing environment which covers certain sections beyond tunnel entrances and exits, is created by multiple factors, such as natural light, artificial light, active or passive lighting signs and marks inside tunnels, light reduction anti-glare facilities, tunnel wall color or decoration and landscape facilities, etc., and is also synthetically affected by light source characteristics such as illuminance, color temperature, and color rendering with their coupling effects. Tunnel light environment is drivers centered, and various light environment characteristics in tunnel influence areas have different effects on drivers’ vision and recognition. In order to smoothen the transition of running tunnel light environments, and then to ensure drivers’ visual performance and driving comfort, it is necessary to divide tunnel lighting area into several zones according to variances of lighting requirements and drivers’ demand in different tunnel locations; doing this will build a comfortable atmosphere for drivers to drive safely through tunnel area with appropriate speed and workload; this also enable drivers to timely obtain requisite information for driving safety in different lighting zones and prevent hidden dangers caused by insufficient visual information.

### 2.2. Necessity to Improve Tunnel Light Environment

Highway tunnel is a very special road traffic operation environment, which is often confronted with large difference between inside and outside tunnel light environment and low luminance level in the interior zone. The particularity of the tunnel environment requires a driver’s higher level of driving skill than that on open roadways, but this is limited by the driver’s ability threshold as a person in the dynamic process in which visual information synthesizing and decision-making are continuously taking place. Vision as a decisive factor of this process is affected by physiological function of human eyes and characteristics of external light source.

For different weather conditions and time periods, tunnel light environment varies a lot from natural light environment. When drivers experience the transition zone between natural and artificial light environment, severe light environment discrepancy will lead to white hole or black hole effects with drivers’ vision then forming a blind vision period. Therefore, it is particularly necessary to set up reasonable and effective lighting conditions in threshold, exit, and interior zones to meet driving needs in different time periods.

On the other hand, although a variety of standards exist for traditional tunnel lighting, there is no mature and stable technology to detect and control artificial light sources, which brings out high cost on energy consumption. As a result, it is quite difficult to effectively describe and accurately obtain visual requirements for driving in a tunnel light environment, this makes it hard to cooperate with driving behaviors. In view of the mutability of natural light source, tunnel light environment must systematically take into account the requirements of drivers’ visual perception, driving safety and comfort, and low energy consumption of artificial light sources.

In the meantime, energy saving should not be at the cost of a lowered safety level of tunnel operation. One widespread adopted method via turning off lights on one side will cause uneven brightness inside tunnels thus forming a Zebra effect, which is prone to causing visual fatigue of drivers. In addition, if locations and angles of lighting fixtures are not reasonably set up, once a glare phenomenon is induced, it will impact drivers’ cognition, judgment and decision on vehicles or obstacles in front, thus bringing risks to driving safety.

In addition, how to monitor and control tunnel light environment, how to evaluate natural and artificial light environments, and what criteria and techniques to apply under different working conditions are all challenging interdisciplinary issues worthy of in-depth study.

In a word, tunnel lighting is targeted to provide a safer, more comfortable and beautiful road environment for different drivers [7]. However, because of the inadequacy of analysis about drivers’ physical and psychological requirements, together with insufficient synthesis on the influence factors of tunnel lighting quality, many tunnel lighting control systems in service fail to balance actual tunnel managements and driving demands, this imbalance will make it hard to meet drivers’ visual comfort, psychological comfort, and convenience of tunnel maintenance simultaneously.

### 2.3. Existing Guidelines for Tunnel Lighting Design

Although there is no uniform tunnel lighting standard globally, International Commission on Illumination (CIE) and most countries adopt luminance based evaluation indexes and specifications; however, standard values are of great difference. Most Chinese tunnel lighting designs refer to luminance based technical specifications as developed countries do. Current Guidelines for Design of Lighting of Highway Tunnels [8] divides a tunnel lighting area into threshold zone, transition zone, interior zone, and exit zone, which are respectively managed with daytime and nighttime patterns. However, only a luminance reduction factor for threshold zone and luminance for other zones are used to evaluate tunnel lighting quality; these evaluation indexes are not adequate. There is no comprehensive analysis on the mechanism of drivers’ visual needs in the tunnel environment, there is no research on the correlation of driving behavior characteristics with different lighting conditions made from different light sources; in addition, technological progress of modern lighting sources has not been objectively and scientifically adopted to monitor and control the lighting environment design, which should be enhanced to improve tunnel driving safety and comfort level.

## 3. Theory and Evaluation Indexes on Tunnel Lighting

CIE suggests tunnel lighting safety be judged by determining whether tested drivers could find target obstacles at a distance not less than a stopping sight distance (SSD) [9]. This method can directly reflect physiological and psychological reactions of drivers under various lighting conditions. If visibility impact could be ignored and drivers could find target obstacles in time outside the SSD while entering or leaving tunnels at specified speed, this light environment should be qualified to ensure driving safety.

As for the relationship of target obstacle and SSD, CIE proposed an evaluation method for tunnel lighting based on optical measurements, a gray cube with side length 20 cm and surface reflection coefficient 0.2 is recommended as the reference to evaluate SSD. Although actual obstacles may be of different size or shape, studies show that this recommended cube still applies to evaluate a variety of lighting environments [10].

### 3.1. Dynamic Visual Recognition Needs

The visual perception and judgment ability of drivers in dynamic driving are obviously different from that in stationary state. According to motor vision psychology, dynamic vision is about 10% to 40% lower than static vision [11]. During dynamic driving, visual observation ability degrades due to decreased visual acuity and narrowed field of vision, and visual cognition is significantly attenuated compared with that under static conditions. Therefore, it is hard to accurately percept and judge abnormal information outside vehicles, thus the risk of driving errors increases. In long tunnels, if mandatory measures could be taken to widen the narrowed field of vision for example changing tunnel landscape, this will prompt drivers to change the field of vision of the fixation point; then, frequency of traffic accidents will be reduced.

Cognitive speed and accuracy not only depend on physiological function of eyes and visual experiences accumulated by brains, but also relate closely to nature of lighting sources in the visual environment. Nature and level of ambient lighting impact drivers’ visual recognition obviously. Luminance enables drivers to sense the change of light and shade from visual environment. When an object is observed under different light source characteristics, seeing the dark from bright needs higher requirements than seeing the bright from dark, and luminance difference has a certain influence on visual recognition of human eyes. If road surface luminance can ensure a driver’s visibility, it will also meet his visual requirements to cognize other information conditions.

Speed and reliability of visual recognition are also affected by brightness contrast or color difference between target and background. For low illumination, this speed is very slow. With the increase of illumination, this accelerates, but this speed does not change significantly when illumination exceeds 1000 lx.

Significant differences exist with drivers’ awareness for daytime and nighttime; normally, night vision is about 1/2 of day vision; this is because the luminance of obstacles decreases when natural light dims rapidly, but drivers’ dark adaptation has not fully built up, weakened contrast between obstacles and ambient background causes drivers’ visual cognition dysfunction [12]. Consequently, night driving relies highly on artificial lighting.

### 3.2. Visual Safety and Comfort Theory

In tunnel entrance and exit area, excessive luminance difference will significantly prolong drivers’ visual adaptation and cause short-term impairment of visual cognitive function, then resultant rise of psychological load will affect driving safety greatly.

Human eyes have different visual perception with various light source characteristics (color temperature, color rendering, light intensity, etc.). In the daytime, if artificial light in a threshold zone is close to natural light, drivers could adapt quickly and feel comfortable. At night, drivers have better adaptability to light sources with low color temperature and feel more comfortable. When foggy, light sources with low color temperature favor drivers to recognize tunnel entrances easily.

In the visible spectrum range, human eyes have different sensitivities to distinct light wavelengths. Tunnel lighting should not focus merely on physical light properties, but also a drivers’ visual system response to light should be cared about, which is also called dynamic visual sensitivity to a visible spectrum [9].

The color of light source depends on the relative energy ratio of different wavelengths it emits. Spectral composition of light source determines not only its color temperature, but also the color rendering effect of illuminated objects. In the bright environment, human eyes are most sensitive to yellow-green light with a wavelength of 555 nm, and in a dark environment, eyes sense acutely blue-green light with a wavelength of 510 nm. The visual state between above two environmental luminances is called intermediate vision, which is basically equivalent to luminance level of general road lighting [13].

When driving inside tunnels, if intervals of road lamps induce zebra crossings with alternating light and shade, and if this alternation ranges within 2.5 Hz to 15 Hz, drivers will feel a stroboscope effect that maybe causes visual discomfort and psychological interference. In addition, glare caused by tunnel lighting possibly disrupts the adaptation of a visual system to surrounding physical space, and then induces visual discomfort or degradation.

### 3.3. Light Source versus Driving Safety and Comfort

#### 3.3.1. Characteristics of Natural Light Source

Characteristics of outside natural light mainly relates to longitude and latitude of the tunnel, solar altitude angle, season, weather, time period, cloud, ground reflection ability, atmospheric transparency, etc. In a specific tunnel area, color temperature of natural light is relatively stable, especially when cloudy or foggy. However, when sunny, color temperature changes gently from low to high and then back to low; peak value usually appears at noon. Outside luminance is basically determined by season, weather, time period, direction of tunnel entrance and surroundings, especially by weather conditions; it changes gently when cloudy (coverage exceeds 0.8) and fluctuates a lot when sunny.

#### 3.3.2. Visual Adaption in the Transition Zone

Ahead of threshold zone, drivers’ eyes have adapted to the luminance of the main field of view within a certain driving distance outside the tunnel. This adaptive luminance is generally considered to be the average luminance measured in the field of view of 20° at SSD in front of the tunnel entrance, which is called L_20_ value [8]. This value directly affects tunnel lighting design and is the reference parameter especially for threshold zone. With the characteristic changes of nature light source, lighting parameters (luminance and color temperature) of artificial light source should change accordingly to ensure drivers’ visual recognition of road conditions ahead.

When inside and outside light environments are of great difference, drivers need some time to adapt to the big change of light intensity. When driving at high speed, sudden illumination change possibly induces black hole or white hole effects. In order to reduce the discomfort caused by this, according to geographical location, environmental conditions, entrance forms, and terrain conditions of the tunnel, different forms of light reduction and anti-glare structures, such as sunshade, shading board, vegetation, pergola [14], etc., can be set up to build a light environment transition zone; this helps very much to couple natural light with artificial light in a friendly manner in the process of visual adaptation.

#### 3.3.3. Characteristics of Artificial Light Sources

Tunnel lighting quality depends heavily on reliable light sources. Selection of tunnel light source must be considered comprehensively and rationally, which is limited by special requirements in tunnel scenario, such as illuminance, photochromic properties (color temperature and color rendering), light efficiency, light flux, light attenuation, lifetime and cost, as well as good visibility even in a smoky atmosphere caused by vehicle emissions. Traditionally incandescent, fluorescent, and high pressure sodium lamps (HPSL) are used for tunnel lighting. At present, light sources for tunnel lighting mainly include fluorescent, HPSL, and LED lamps; incandescent lamps are no longer installed. Table 2 compares the technical parameters for HPSL and LED lamps.

This comparison between HPSL and LED indicates LED is more suitable for tunnel lighting; this conforms to the target of energy saving and emission reduction, also assuring drivers’ safety and comfort of driving cognition.

### 3.4. Evaluation Indexes of Light Environment in Highway Tunnels

Global research on tunnel light environment mainly builds on tunnel luminance condition; there are three primary methods to evaluate lighting quality via visual effect, namely, visual performance used for short time recognition, visual fatigue used for long duration visual tasks, and subjective evaluation method based on visual psychological satisfaction. In this case, tunnel light environment is usually subdivided into access zone, threshold zone, transition zone, interior zone, and exit zone [8,15], as Figure 5.

Research shows that drivers’ ability of visual recognition in tunnel lighting zones not only relates to road surface luminance and uniformity of road surface luminance but is affected by characteristics of light source and luminance difference between inside and outside of tunnels. However, there are no relevant regulations on photochromic properties and spectral distribution of light sources in current specifications. Hence, tunnel lighting design firstly needs to abide by the requirements in current lighting guidelines, and it is also desirable to study characteristics of light sources systematically and apply them reasonably; this will contribute to balancing favorable lighting effects and energy saving.

#### 3.4.1. Color Temperature and Color Rendering Index

Warm or cold feelings of light color relates to human physiological and psychological effects. Appropriate color temperature aids in balancing the functions of the central nervous system and autonomic nervous system, and to keep nerves in a relaxed state. Otherwise, it may lead to dysfunction of the central nervous system and even disturb the body’s natural balance [16]. Some experiments indicate that different spectral compositions of light source and adaptation level of human eyes influence visual response time greatly, and the radiation spectrum containing more blue and green light will shorten this response time.

Relative spectral power distribution of various wavelengths in light source determines its color rendering index (CRI). Light sources with continuous spectral distribution, wide spectral coverage, and spectral energy characteristics close to natural light have high CRI. CRI directly affects if drivers could recognize colored objects correctly. In a CRI test with small targets, Yamamoto found that, under intermediate visual range, different spectral composition of light sources influences if small colored targets could be correctly recognized a lot. On the premise of meeting the same correct recognition rate of small targets, tunnel light sources with high CRI and high content of short-wave components are more conducive to improving visual recognition [17].

Traditionally, tunnel lighting mainly adopted HPSL, and modern tunnel designs tend to choose LED due to its adjustable color temperature and high CRI. Because various combinations of color temperature and CRI have different influences on drivers’ ability of visual recognition, they are important indexes to characterize tunnel light sources.

#### 3.4.2. Luminance Reduction Factor

While driving inside from the outside, visual recognition is impacted not only by road surface luminance of the threshold zone, but by an outside luminance level. Luminance difference between inside and outside is the main index affecting driving behavior characteristics at the threshold zone. Thus, designing appropriate luminance transition between threshold zone and outside is vital to improve drivers’ visual recognition. Luminance reduction factor *K* is defined to describe this difference between threshold zone and outside:*K* = *L_th_*/*L*_20_(1)

Here, *L_th_* is the average road surface luminance at threshold zone, and *L*_20_ is the average luminance at a safe stopping sight distance with 20° field of view opposite the tunnel entrance.

Similar to the absence of uniform luminance standards, most countries and academic organizations have no uniform provisions on luminance reduction factor in the threshold zone. Actual reduction factor can be determined appropriately through experiments according to characteristics of natural light source in the access zone during daytime, design speed, traffic volume, and required threshold to satisfy driving safety and comfort, so as to harmonize driving safety with energy saving.

Cognitive and behavioral characteristics of drivers at tunnel entrances and exits indicate that, when driving in the environment with luminance difference between inside and outside at night, if this difference exceeds a certain threshold impacting drivers’ visual cognition and even producing a white hole or black hole effect, at this moment, the most common preventive measure drivers tend to take to alleviate the unsafety and discomfort is changing vehicles’ speed [18], but this speed shift will further worsen the speed standard deviation and variance among different vehicles, which have a positive correlation with traffic accident rates [15,19]. Therefore, luminance difference between inside and outside is an important index to characterize the validity of visual cognition for a light environment at the tunnel entrance and exit area.

#### 3.4.3. Road Surface Luminance

Drivers’ visual perception is closely related to luminance of light environment, but there is a luminance threshold for the self-tuning ability of drivers’ visual systems. When driving on the road, it is road surface luminance that drivers feel more real rather than illumination of light source. The luminance method evaluates drivers’ visual ability by estimating the luminous flux of light source reflected from road surface into drivers’ eyes. Static experiment shows that drivers’ ability of visual recognition varies for the same light source with the same photochromic properties but different road surface luminance as Table 3.

Road surface luminance is the actual perception of light intensity reflected from road surface to drivers’ eyes; it directly affects drivers’ recognition on road obstacles, and road situations can not be clearly recognized in case of insufficient luminance. Therefore, this index is a globally accepted indicator to evaluate the validity of visual recognition in a tunnel light environment, and its minimum is prescribed to 1.0 cd/m^2^.

#### 3.4.4. Uniformity of Road Surface Luminance

Because road surface luminance is from the reflected illuminance emitted by an overhead artificial point light source that projects and superimposes on the ground, if uneven superposition or reflection exists, non-uniform dark spots or shadows will appear, then resultant wavy feeling will endanger driving safety more or less. Onaygil and Sennin discussed the influence that luminance uniformity acts on visibility level in road lighting design and took visibility level as an indicator to measure driving comfort and safety [20].

Flicker frequency that resulted from tunnel lamps installation should not be within 2.5–15 Hz to avoid causing visual discomfort and psychological interference to drivers, which is based on the fact that impact on people is negligible when flicker frequency is lower than 2.5 Hz or higher than 15 Hz. The spacing between lamps is larger in an interior zone than that in the threshold zone, so it is difficult to ensure good uniformity in an interior zone. In some areas of the interior zone, there are maybe some dark bands with low luminance, if bright and dark bands appear repeatedly, this resultant zebra effect will disturb drivers’ visual recognition for ahead road conditions. This non-uniformity is to induce negative psychological reactions making drivers feel irritable.

In terms of lighting effect, given the same tunnel environment and lamps, road surface illumination produced by staggered arrangement is slightly better than that by symmetrical arrangement, and total uniformity of road surface luminance is higher than that by symmetrical arrangement.

Stepless dimming advantage of LED light source makes it unnecessary to turn off partially to reduce energy consumption; instead, it is feasible to finely adjust the output power of lamps according to current traffic flow and maintenance factor of lamps, which could reduce energy consumption and ensure the uniformity of road surface luminance.

All of the above analysis shows that uniformity of road surface luminance is an important index to characterize tunnel light environment.

#### 3.4.5. Visual Cognition Distance

When driving in tunnel area, drivers need to clearly recognize the linear conditions, road conditions, signs and markings, vehicles information, obstacles, and landscape conditions within ahead safe distance. Only by ensuring enough safe visual distance can drivers finish the complete process consisting of cognition, judgment, decision-making, and operation to ensure driving safety. In this study, as suggested by CIE, SSD is used to evaluate drivers’ visual recognition process to determine if target obstacles can be found at a distance not less than SSD, on which to judge the safety of a tunnel light environment.

#### 3.4.6. Other Indexes

Besides the above indexes for tunnel light environment, in this study, the ways below are used to indirectly evaluate the quality of a tunnel lighting control system:(1)With help of actual traffic flow data at peak hours, the average speed of traffic flow, standard deviation, or variance of vehicle speeds are statistically analyzed; this method is able to indirectly indicate the smoothness of tunnel traffic flow at designed speed.(2)It can be regarded as another subjective evaluation method of lighting quality through questionnaire collecting real feedback of drivers on tunnel lighting or experience of safety and comfort during tunnel driving.(3)Based on data of lighting energy consumption, horizontal comparison results with similar tunnels (similar in geographical location, length, traffic flow, and other properties) can be thought of as an objective index to assess the economy of tunnel lighting.

## 4. Design of Lighting Control System

### 4.1. Selection of Light Source and Control Mode

Research of tunnel lighting control mainly includes two aspects, overall architecture of lighting system and control mode of lighting lamps. The lighting control system has gone through several stages, namely centralized control, distributed control, and fieldbus control. Currently, fieldbus control built on industrial Ethernet dominates most newly constructed tunnels. Control mode of tunnel lamps includes roughly three categories, artificial control, time sequence control, and presently the mainstream intelligent control. Meanwhile, stepless dimming has replaced traditional logic switching and becomes the primary control mode because LED lamps can be continuously dimmed within a full range.

With the advance of LED lighting theory and manufacturing technique, LED lighting is being widely used in bridge and tunnel construction. Olijinyk studied the advantages of LED on energy saving as tunnel lighting sources used by TIR Systems Inc. [21]. Guo et al. combined stepless dimming mode with an LED light source, and achieved 57% energy savings through transformation of traditional extensive lighting control [22]. Wang et al. replaced traditional HPSLs with LEDs, and reformed lighting control from depending only on luminance to variable color temperature together with stepless dimming, which brought rewards of 18% energy saving [23]. Zhang et al. designed an automatic lighting control mode (i.e., tunnel lamps switch on when car enters and off after leaves) via PID closed-loop feedback and road surface luminance measurement with image processing [24]. Today, on the Chinese Hongkong-Zhuhai-Macau Bridge, LED lamps and lighting control systems are serving the lighting project, which includes the main project of one 22.9 km bridge connected by one 6.7 km tunnel and branch project composed of multiple artificial islands [25].

### 4.2. Topology of System and Components

As Figure 6 shows, full consideration is taken on the advantages of distributed system and hierarchical management model, from the perspective of logic functions and control flows, this system is designed into two levels; namely, top-level tunnel system manager and tunnel onsite manager will implement system designed functions as Table 4.

Tunnel System Manager deployed in a central control room includes a Master Control Module of this system software, the lighting APP module responsible for running an intelligent dimming strategy, and the Data Server module in charge of transferring data and control instructions amongst different software components. Lighting APP receives periodic data generated by perception layer and transferred by Data Server, such as lighting condition, tunnel facilities, traffic flow, etc. and then decides next-cycle light adjustment parameters, and output control commands through Data Server to Tunnel Onsite Managers.

Tunnel Onsite Manager mainly consists of Adaptive Lighting Controller, Light Environment Sensing Unit, and Local Lighting Module. The Adaptive Lighting Controller mainly realizes two functions. Firstly, it provides the upper Lighting APP module with light sensing data (outside color temperature and luminance) sampled by light sensors located outside of the tunnel entrance. In addition, it addresses lighting control instructions output by upper Lighting APP to corresponding Local Lighting Modules. The Local Lighting Module takes charge of interpreting lighting control signals translated by Adaptive Lighting Controller and converting them into appropriate voltage values and proportions for yellow and white light, respectively. Then, it sets the power supply modules of specific lamp groups covered by the Local Lighting Module; the lamp groups will produce expected luminance and color temperature. In the current system, Adaptive Lighting Controller and Local Lighting Module can be addressed individually, and lamps are controlled in groups. In addition, if needed, it is also feasible and easily accomplished to adjust lighting parameters for a single lamp in fine granularity by assigning a unique address to each lamp.

### 4.3. Software Framework Based on System Requirements

As Figure 7 shows, this system consists of four layers, namely, Perception, Transport, Service Management, and Application; this structure is flexible and requirements orientated as Table 5.

Perception layer mainly acquires and normalizes all kinds of data from multiple heterogeneous sensors deployed in tunnels, such as light sensor, vehicle detector, ventilator, electricity meter, water pump and fire control facilities, video and audio devices, etc.; these data will feed into the dimming policies residing in Lighting APP.

In order to suit the diversity of perception devices, the transport layer supports several access media, namely wireless cellular, optical network, RS485, and WLAN, providing transparent and reliable links to exchange data between lower and upper layers.

As a key component of this control system, service management layer consists of a data plane in charge of data storage and transfer, traffic plane for routine operations, control plane to handle control commands for different layers and monitor status of lower facilities, policy library to execute lighting strategies, and service library for extra user demands.

The application layer provides operating personnel with friendly GUI to monitor real-time conditions of tunnel environment via two subsystems. An intelligent monitoring subsystem presents the running states of tunnel facilities, traffic flow, and dimming strategy, tracks if lighting APP effectively responds to changes of outside light conditions, and diagnoses if this system performs well to meet driving safety and comfort requirements. Meanwhile, the operation and maintenance subsystem offers conveniences for manual calibration and correction of system deviation.

Parallel with these four layers, Diagnostics and Maintenance API is reserved to facilitate online detection and trouble-shooting with all lighting equipment and software components during system integration and after deployment.

### 4.4. Software Process of Lighting APP

The process of intelligent lighting software mainly means how lighting APP outputs specific lighting control commands. This software aims to adjust a tunnel light environment for different design speeds of 40 km/h, 60 km/h, 80 km/h, 100 km/h, 120 km/h, etc., in the meantime, other multiple factors, namely weather, traffic volume, color temperature, and luminance, are considered together. Friendly coupling of internal and external light environments will be achieved through regulating artificial lighting parameters.

Lighting APP is adaptive for four kinds of weather changes outside tunnels:(1)Regulation under normal weather conditions;(2)Regulation under abnormal weather conditions;(3)Regulation when normal weather changes to abnormal weather conditions;(4)Regulation when abnormal weather changes to normal weather conditions.

Each of these scenarios matches its own criteria for color temperature and luminance conversion; lighting adjustment is executed in smaller steps of color temperature and luminance in turn without introducing significant transition of visual comfort. Figure 8 is used to exemplify the basic algorithm and process corresponding to design speed of 80 km/h in the daytime at the threshold zone; this pattern also suits for other design speeds.

In this figure, acronyms and legends as Table 6 are used to denote decision conditions.

The major principle of this lighting strategy is to avoid significant transition of visual comfort, so adjustment of lighting parameters is executed in steps as small as possible for both internal color temperature and luminance, which will make drivers feel no visible change of lighting conditions,

(1)If color temperature will transition across a span larger than 500 K or luminance transition is greater than 2 cd/m^2^, lighting adjustment should be finished in multiple steps for color temperature or luminance;(2)The number of color temperature adjustment steps (*StepCT*) equals the biggest integer of color temperature span,

*StepCT* = *Ceiling*[*Abs*(*M* − *m*)/500];(2)

(3)The number of luminance adjustment steps (*StepL*) equals the biggest integer of luminance span,

*StepL* = *Ceiling*[*Abs*(*A*1 − *A*2)/2];(3)

(4)If both color temperature and luminance are required to be adjusted in single cycle, color temperature adjustment has higher priority than luminance change.

### 4.5. Reliability Assurance Aided by FMEA

Failure Mode and Effects Analysis (FMEA) is a systematic procedure to analyze potential failure modes and causes, and their impacts on system performance. It is both a program to assure the quality of system development and a preventive thinking model [26,27,28]. FMEA can be qualitative or quantitative [29,30], and is often the vital step to study the reliability of a specific system. In this study, before the lighting control system is officially put into operation, system analysis and design aided by FMEA can find potential failures as early as possible and reduce failure risks at lower costs, and eventually enable this system to be reliable and robust to meet required design criteria with a high confidence level.

Based on these system requirements, overall architecture, and network topology, there are mainly the following failure modes:(1)Loss of function;(2)Functional degradation;(3)Intermittent functional failure;(4)Function delay;(5)Unexpected functions.

The FMEA method determines three parameters for each failure mode caused by a single cause according to predefined grading standards, namely severity (S), occurrence (O) and detection (D), and each ranges from 1 to 10 as Table 7, Table 8 and Table 9, respectively.

For each failure mode, Risk Sequence Number (RPN) is calculated as Equation (4) by referring to above Table 7, Table 8 and Table 9, namely product of *S*, *O*, and *D* that ranges from 1 to 1000. In addition, if this *RPN* exceeds 100, measures should be taken to reduce this *RPN*:*RPN* = *S* × *O* × *D*(4)

With this process iterates, when *RPN* of each failure mode is small enough (normally less than 20) and its severity is less than 7, failure risk of whole system is fairly low and its reliability is increasing. During this process, preventive and optimization measures need to be filed to provide reference for subsequent development, testing, and maintenance.

## 5. Application and Evaluation of Lighting Control System

### 5.1. Application

YanChong highway is the major channel connecting Beijing and Chongli competition areas for 2022 Beijing Winter Olympic Games. It is also listed in the first batch of green road constructions and typical demonstration projects approved by the Chinese Ministry of Transport. As one of the important control projects of YanChong highway, construction and operation of Songshan tunnel should firstly consider the potential factors related to traffic accidents due to its 9.2 km total length; another concern is with ecological environment since it is adjacent to one national nature reserve. Besides these requirements, driving safety and low-carbon operation should be given more attention. As of today, one intelligent lighting control system conforming to these premises has been deployed after the acceptance test and runs normally with continuous supervision of lighting quality.

As Table 10 shows, a light environment sensing unit at the tunnel entrance mainly consists of a luminance detector and a color temperature thermometer, which are used to sense and collect real-time light conditions and flow the transformed light data into the onsite adaptive lighting controller. This controller is essentially one highly reliable DA-1000 industrial computer driven by the CentOS system; it collects light data and manages local lighting modules. A local lighting module is evolved from CC09TA with adjustable voltages ranging from 0–10 V; it parses control commands from adaptive lighting controller and outputs well-proportioned voltages via RS485 bus to drive the corresponding LED lamp group to emit white and yellow lights separately matching with target color temperature and luminance. These LED lamps with 70 w rated power are voltage adjustable for white and yellow lights individually within 0–10 v, and color temperature ranges 3000–7000 K.

Master control and lighting APP modules reside in upper computers located in a tunnel station; they work as a top-level coordinator for all components inside this system.

### 5.2. Evaluation

Since its deployment, this system has experienced a variety of working conditions in different seasons and weather, and the research team has conducted a mass of field tests and collected vast amounts of running data.

After periodic load tests and field verification, this system works stably:(1)Average road surface luminance and uniformity of road surface luminance meet expected lighting specifications;(2)Color temperature and luminance of LED lamps alter as intelligent lighting strategy requires;(3)Artificial light environment of each tunnel zone offers adequate visual cognition distance for driving safety and comfort;(4)Inside this tunnel, traffic flow runs smoothly and there is no obvious speed variability;(5)Feedback from actual drivers through this area indicates that this lighting quality satisfies visual performance, and little visual fatigue is induced.

Figure 9 and Figure 10 are used to exemplify how color temperature and luminance in a threshold zone vary on a 10-min cycle in normal weather with outside light data for consecutive 12 h from 6:00 A.M. to 6:00 P.M.

In Figure 9, there are twice the transitions between 6000 K and 3500 K, 3500 K lasts roughly in the midday duration. During this time window, drivers are prone to struggle with visual fatigue; this transition of color temperature between threshold zone and open space helps a lot with stimulating drivers’ excitement and making them concentrate on driving carefully, eventually keeping the driving process safe and smooth in the tunnel area.

Figure 10 shows that the luminance trend in the threshold zone is basically consistent with that of the outside tunnel, which contributes a lot to reducing visual discomfort caused by a black hole effect while approaching the threshold zone.

However, this luminance curve shifts slightly at the moment when color temperature transitions between 6000 K and 3500 K; this is because different L_20_ luminance reduction factors are assigned in the lighting strategy to make up color temperature fluctuation in the daytime under normal weather. In the meantime, since target lighting parameters are possibly adjusted in multiple steps, a large gradient change of luminance or color temperature is subdivided into several time slices with smaller granularity in each regulation period, which effectively weakens the harm that large gradient luminance or color temperature jitter may cause.

## 6. Conclusions

Different from many existing lighting control systems, technical means are explored in this system to alleviate visual discomfort from the standpoint of driving safety and comfort requirements in the tunnel light environment, which is theoretically founded on the correlation research of variable color temperature technology of an LED lighting source with human factor engineering. After its deployment with an actual highway tunnel, this system is serving tunnel users with high lighting quality, which is verified by testing results and actual operation data; this system is adaptive with high reliability, it contributes a lot to optimizing lighting conditions and reducing the fluctuation of driving workload; as a result, driving safety level is somewhat enhanced in return. This indicates that this LED lighting control system is quite practical for promoting visual effects and reducing traffic risks in long and extra-long tunnels.

In view of the potential out-of-limit deviations between actual lighting performance of LED lamps and ideal parameters calculated by an intelligent lighting algorithm, in the long run, it is highly preferred to introduce a feedback link of closed loop control to offset the deficiency of current open loop control. In addition, if an artificial intelligence algorithm could be used to deeply mine the mass data accumulated by this system, it will be of great benefit to optimize the emergency plan and time sequence control mode of tunnel lighting.

## Figures and Tables

**Figure 1 ijerph-19-08517-f001:**
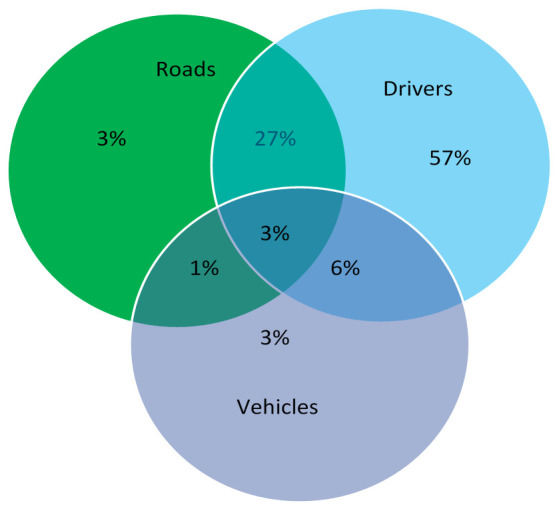
Causes of traffic accidents in America.

**Figure 2 ijerph-19-08517-f002:**
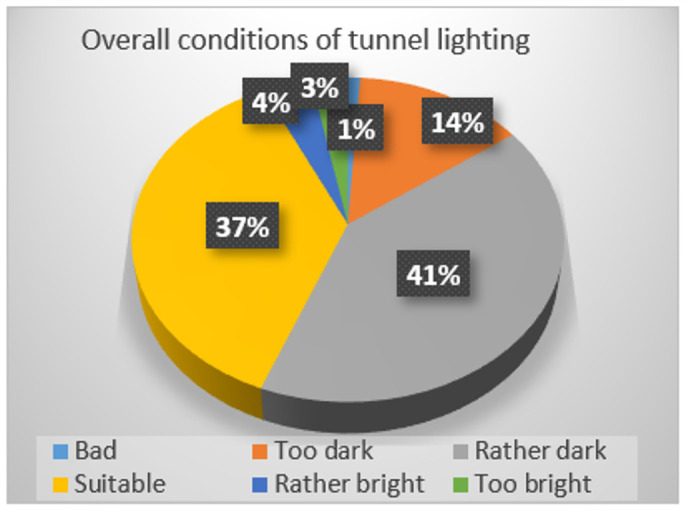
Overall conditions of tunnel lighting.

**Figure 3 ijerph-19-08517-f003:**
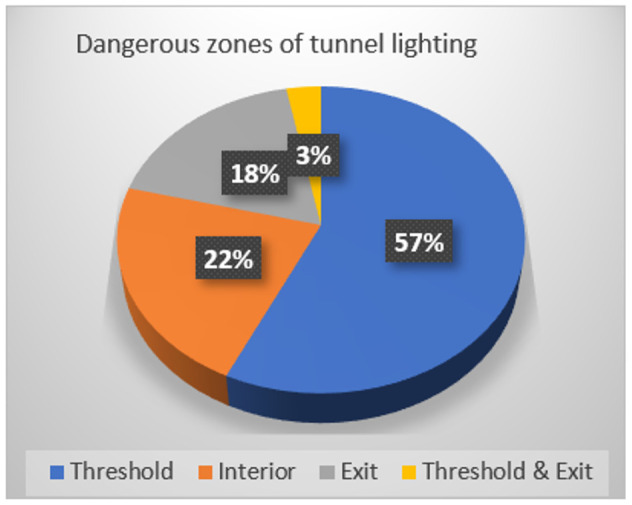
Dangerous zones of tunnel lighting.

**Figure 4 ijerph-19-08517-f004:**
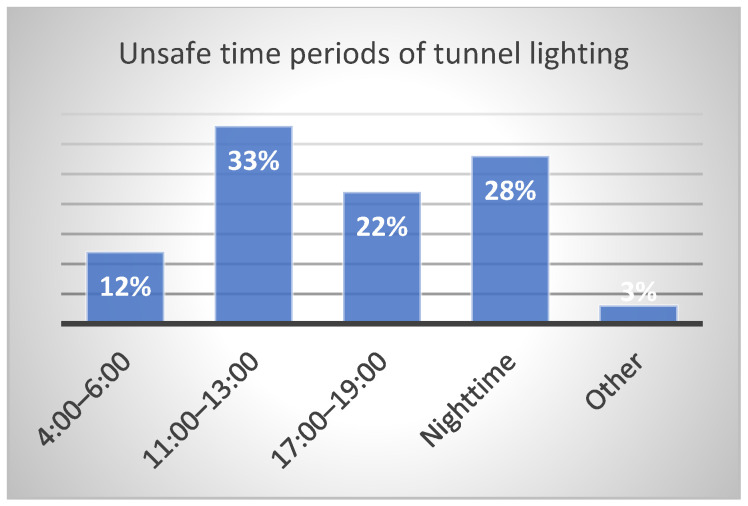
Unsafe time periods of tunnel lighting.

**Figure 5 ijerph-19-08517-f005:**
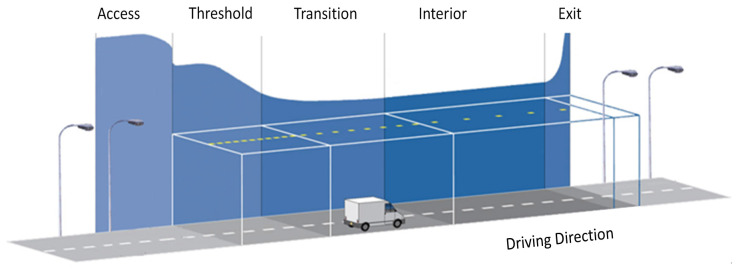
Tunnel zones for light environment.

**Figure 6 ijerph-19-08517-f006:**
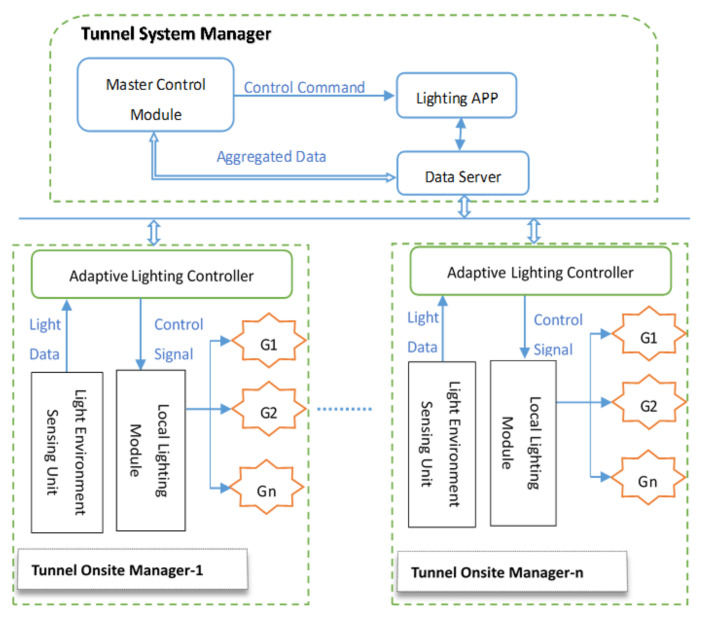
Network topology of the tunnel lighting control system.

**Figure 7 ijerph-19-08517-f007:**
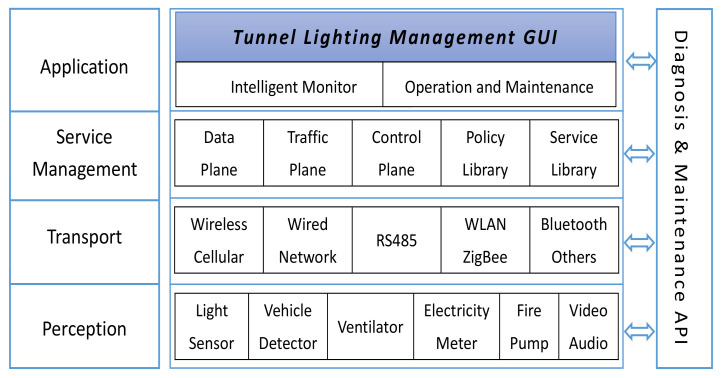
Overall architecture of the tunnel lighting control system.

**Figure 8 ijerph-19-08517-f008:**
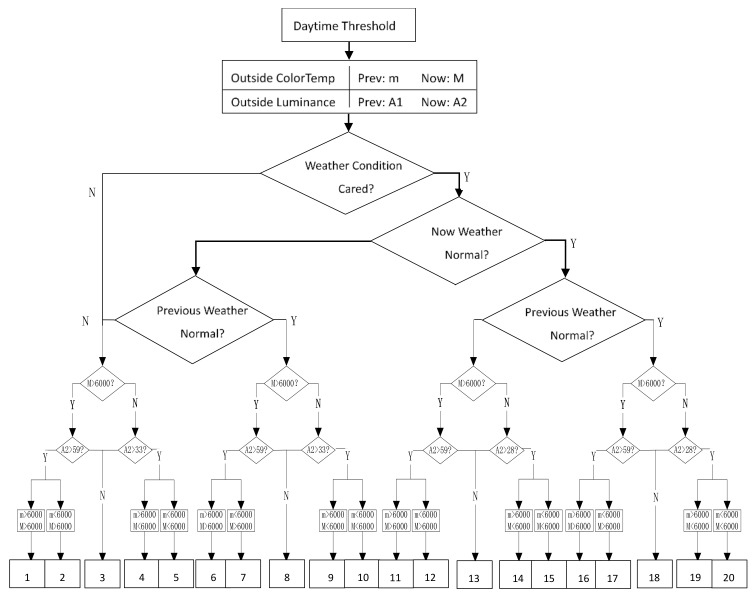
Flowchart of Lighting APP at 80 km/h.

**Figure 9 ijerph-19-08517-f009:**
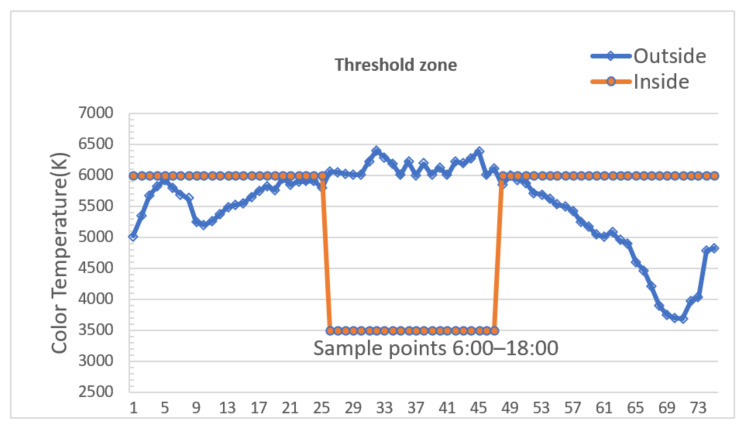
Color temperature curve in a threshold zone.

**Figure 10 ijerph-19-08517-f010:**
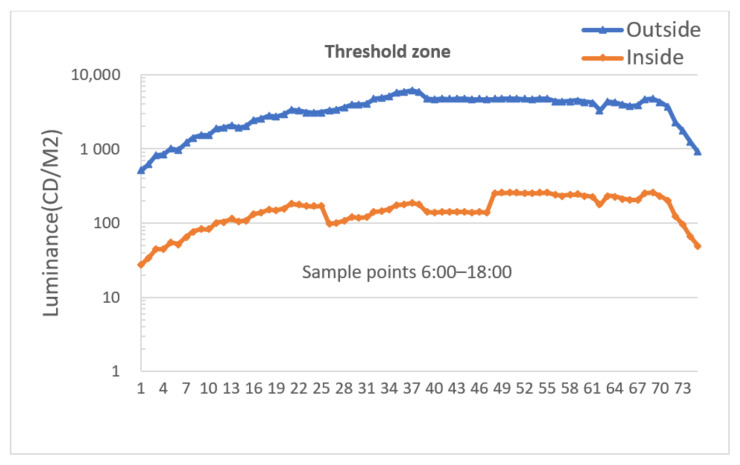
Luminance curve in a threshold zone.

**Table 1 ijerph-19-08517-t001:** Progress of tunnel construction in China (2011–2020).

	2011	2012	2013	2014	2015	2016	2017	2018	2019	2020
Total	8522	10,022	11,359	12,404	14,006	15,181	16,229	17,738	19,067	21,316
Long	1504	1944	2303	2623	3138	3520	3841	4315	4784	5541
Extra-long	326	441	562	626	744	815	902	1058	1175	1394

**Table 2 ijerph-19-08517-t002:** Comparison of HPSL and LED.

	HPSL	LED
**Service Life**	Short service life ^1^	Long service life ^1^
**Color Rendering**	Low CRI, bad restoration to original color.	High CRI, restoration to original color with minor distortion.
**Color Temperature**	Narrow range (2000–3000 K).	Wide range (3000–6500 K).
**Utilization Factor**	Omnidirectional luminescence, 60–80% luminous flux reaches the ground.	Directional luminescence, More than 85% of the luminous flux reaches the ground.
**Others**	Pollutants (Mercury, lead), Slow startup and restart.	-

Note: ^1^ Compared with the same criterion of light attenuation (70%) or damage ratio.

**Table 3 ijerph-19-08517-t003:** Visual recognition under different luminance levels.

Color Temperature: 5700 K, CRI: 70								
**Luminance (cd/m^2^)**	5	4.5	4	3.5	3	2.5	2	1.5
**Number of frames**	9.60	9.60	11.10	13.00	13.38	12.63	13.50	17.57
**Found times**	10	10	10	8	8	8	8	7
**Ratio**	100%	100%	100%	100%	100%	100%	100%	87.50%

**Table 4 ijerph-19-08517-t004:** Functions and components of the lighting control system.

	Master Control Module	Fulfill Top-Level Management of This Lighting Control System
**Tunnel System Manager**	Lighting APP	Module in which predefined intelligent lighting policies resides. Real-time response to data from Light Environment Sensing Unit. Assembly Lighting Adjustment Command.
	Data Server	Transfer data and control commands among modules. Databases in charge of tunnel management jobs.
**Tunnel Onsite Manager**	Adaptive Lighting Controller	Collect light data from Light Environment Sensing Unit, and transfer this data to Tunnel System Manager. Parse Lighting Adjustment Command from Lighting App, and transfer to Local Lighting Module in the form of Control Signal.
	Light Environment Sensing Unit	Sense outside tunnel light environment. Normalize Light Data (outside color temperature and luminance).
	Local Lighting Module	Parse Control Signal from Adaptive Lighting Controller. Distribute appropriate voltage values to covered LED lamps.

**Table 5 ijerph-19-08517-t005:** System layering and assignment of system requirements.

**Layer-4**	Application	Tunnel Lighting Management GUI Real-time monitoring of tunnel light environment. Ease of Software upgrade and maintenance.	Diagnostics and Maintenance API, Troubleshooting ports reserved for each layer. Online upgrade and maintenance for any modules.
**Layer-3**	Service Management	Data Plane for storage and transfer. Traffic Plane for routine tunnel lighting operations. Control Plane to deliver lighting control commands and status indications of lower facilities. Policy Library to carry out intelligent lighting strategy. Service Library for extra demands and extensibility.	
**Layer-2**	Transport	Diverse and highly reliable communication links and media to relay sensing data and control flow to and from upper layers.	
**Layer-1**	Perception	Sample and normalize sensing data from multiple sensors.	

**Table 6 ijerph-19-08517-t006:** Notes and legends used in Figure 8.

**A2 > 59**	If outside luminance is greater than threshold 59 cd/m^2^, luminance transition will take place. Similar to A2 > 33, A2 > 28, etc.
**M > 6000**	If now outside color temperature is greater than 6000 K, similar to M < 6000. Different color temperate transition pattern, for example m > 6000 and M < 6000, different dimming strategy applies.
**Table 1–20**	Each table (from 1 to 20) corresponds to one separate dimming strategy. Target light environment parameters are decided by the tuple (weather, outside light condition, tunnel design speed, traffic volume).

**Table 7 ijerph-19-08517-t007:** Evaluation criteria for FMEA severity (S).

Severity	Evaluation Criteria	Impact
10	All LED lamps within control area go out with no warning.	critical
9	Whole system or some modules are loss of basic functions and cannot be restored.	
8	All LED lamps within control area flicker on and off with no warning.	very serious
7	Abnormal optical sensing data (color temperature or luminance) in a light environment sensing unit.	serious
6	In a certain lighting zone, LED lamps lose control, luminance, and color temperature cannot be adjusted or obviously lagged behind.	
5	In a certain lighting zone, LED lamps cannot be ideally controlled, and luminance or color temperature cannot be adjusted.	medium
4	In a certain lighting zone, luminance or color temperature of LED lamps are inconsistent with the preset dimming strategy.	
3	System functions normally, its performance meets design requirements with occasional status prompt.	low
2	System and each module function well as expected with internal prompt messages logged.	
1	There are no discernible anomalies.	very low

**Table 8 ijerph-19-08517-t008:** Evaluation criteria for FMEA occurrence (O).

Occurrence	Evaluation Criteria	Failure Probability
10	More than 100 times for every 1000 samples.	extremely frequent
9	80 times for every 1000 samples.	
8	50 times for every 1000 samples.	frequent
7	20 times for every 1000 samples.	
6	10 times for every 1000 samples.	occasional
5	5 times for every 1000 samples.	
4	Once for every 1000 samples.	
3	Once for every 2000 samples.	low
2	Once for every 10,000 samples.	
1	Less than once for every 10,000 samples.	Almost never

**Table 9 ijerph-19-08517-t009:** Evaluation criteria for detection (D).

Detection	Evaluation Criteria	Chance
10	Unable to design probe methods.	no chance
9	Weak probe ability, hard to simulate real conditions.	basically undetectable
8	This mode can only be simulated.	extremely low
7	Mode and mechanism can be located empirically.	very low
6	No obvious dimming effect, more than 1 failure mode can be located with testing equipment.	low
5	No obvious dimming effect, 1 failure mode can be located with testing equipment.	general
4	Mode and mechanism can be deduced depending on predefined lighting strategy.	a bit high
3	Obvious inconformity between dimming effect and lighting strategy, more than 1 failure mode can be accurately located.	high
2	Inconformity between dimming effect and lighting strategy, 1 failure mode can be accurately located.	very high
1	Failure does not occur due to leverage of previous high-quality software or hardware modules.	needless

**Table 10 ijerph-19-08517-t010:** Main functional components and hardware entities.

**Light Environment Sensing Unit**	light luminance detector color temperature thermometer
**Adaptive Lighting Controller**	industrial computer: DA-1000 operating system: CentOS
**Local Lighting Module**	centralized controller: CC09TA regulation voltage: 0–10 v
**LED Lamps**	rated power: 70 w color temperature: 3000–7000 K CRI: 70
**RS485 wires**	remote control to LED lamps
**Upper computers in tunnel station**	host computers shared by Master Control Module, Lighting APP, and Data Server

## Data Availability

Data generated in this study are available upon request.

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
