# Peer review of "Traffic Safety Improvement via Optimizing Light Environment in Highway Tunnels"

_ijerph, 2022, doi:10.3390/ijerph19148517_

Round 1
Reviewer 1 Report
The paper deals with Traffic Safety Improvement via Optimizing Light Environment in Highway Tunnels by means of An intelligent lighting control system designed with multiple influence factors.
Some improvements should be made to the text:
"Driving in tunnel area depends more heavily on light conditions than that on open road-ways": Confusing phrase.
Adjust numbering of the titles of the sections, Improve characteristics of the study carried out related to figures 2, 3 and 4.
Improve quality and information of these figures. Include references in text when it has not been done (i.e. section 2.3.2)
Complete the explanation of section 4.2 and graphs 9 and 10.
Complete conclusions (section 5) specifying details of the study.
Fix bugs in References
Reviewer 2 Report
page 2/19, time is in 24 clock hours, not am/pm
page 3/19, in figure 3, the word should be 'Threshold'
Minor grammatical errors throughout. I recommend an English proofreader to go over this paper.
Overall: a good paper.
Reviewer 3 Report
- The subject of this scientific article is traffic Safety Improvement via Optimizing Light Environment in Highway Tunnels .
- Generally, it can be said that this article fulfils the criteria defined for publication of the scientific-research article.
- Structure of this article is well-arranged and with all the important items.
- It is necessary to emphasize an important contribution of the article, which integrates a topic of the advanced technology .
- My opinion is that this scientific article is interesting and useful within the investigation area of vehicle emissions.
- I recommend to publish this article after implementation of my following comments and suggestions:
1. It is necessary to check the stylistic aspect of the article, some of the sentences are not finished or are without a sense.
2. Readability of fig 8 is insufficient.
Round 2
Reviewer 1 Report
The suggested corrections have been made, I recommend acceptance of the manuscript in its present form.